# Molecular Characterization of Novel x-Type HMW Glutenin Subunit 1B × 6.5 in Wheat

**DOI:** 10.3390/plants10102108

**Published:** 2021-10-05

**Authors:** Tímea Kuťka Hlozáková, Zdenka Gálová, Svetlana Šliková, Leona Leišová-Svobodová, Jana Beinhauer, Filip Dyčka, Marek Šebela, Erika Zetochová, Edita Gregová

**Affiliations:** 1Faculty of Biotechnology and Food Science, Slovak University of Agriculture, Tr. A. Hlinku 2, 949 76 Nitra, Slovakia; timea.kutka.hlozakova@gmail.com (T.K.H.); zdenka.galova@uniag.sk (Z.G.); 2National Agriculture and Food Centre, Research Institute of Plant Production, Bratislavská Cesta 122, 921 01 Piešťany, Slovakia; svetlana.slikova@nppc.sk (S.Š.); erika.zetochova@nppc.sk (E.Z.); 3Crop Research Institute, Drnovská 507, 161 06 Praha, Czech Republic; leisova@vurv.cz; 4Centre of the Region Haná for Biotechnological and Agricultural Research, Faculty of Science, Palacký University, Šlechtitelu 27, 783 71 Olomouc, Czech Republic; jana.beinhauer@seznam.cz (J.B.); fdycka@prf.jcu.cz (F.D.); marek.sebela@upol.cz (M.Š.)

**Keywords:** wheat, HMW glutenin, 1-D electrophoresis, 2-D electrophoresis, mass spectrometry

## Abstract

A novel high molecular weight glutenin subunit encoded by the Glu-1B locus was identified in the French genotype Bagou, which we named 1B × 6.5. This subunit differed in SDS-PAGE from well-known 1B × 6 and 1B × 7 subunits, which are also encoded at this locus. Subunit 1B × 6.5 has a theoretical molecular weight of 88,322.83 Da, which is more mobile than 1B × 6 subunit, and isoelectric point (pI) of about 8.7, which is lower than that for 1B × 6 subunit. The specific primers were designed to amplify and sequence 2476 bp of the Glu-1B locus from genotype Bagou. A high level of similarity was found between the sequence encoding 1B × 6.5 and other x-type encoding alleles of this locus.

## 1. Introduction

Wheat (*Triticum aestivum* L.) has the significant role in human nutrition as the staple food for 40% of the world’s population. Among cereals, wheat flour has the unique ability to form dough. These properties are mainly determined by seed storage proteins which are converted into gluten complex. Glutenins confer dough elasticity and gliadins dough extensibility essential for bread-making quality [1]. Glutenins are classed as high molecular weight (HMW) encoded at *Glu-1* loci and low molecular weight (LMW) encoded at *Glu-3* loci. HMW glutenin subunits are further subdivided into high M_r_ x-type with 80–88 kDa and low M_r_ y-type with 67–73 kDa subunits [2]. Wheat flour consists of 10% of glutenin protein from which only 0.75–1.25% belong to HMW glutenin subunits. This relatively small amount, however, has been found to have an effect on flour quality much greater than suggested by its proportion [3]. Usually one HWM-GS is coded at loci on chromosome 1A, one or two at loci on chromosome 1B and also two on chromosome 1D. It is well known that A × 2* and D × 5 + Dy10 is associated with good bread-making quality, especially dough strength, while D × 2 + Dy12 is associated with poor quality. The highest polymorphism of HMW-GS is regularly detected on 1B chromosome [4,5,6,7]. Nowadays, the most challenging task for wheat breeders is not only to increase grain yield [8] but also to improve the grain quality for end products. In this paper we describe the evidence of new subunit 1B × 6.5 coded by locus *Glu-1B* in the French wheat genotype Bagou. Its originality was proved using SDS-PAGE, 2D electrophoresis and MALDI-TOF-MS protein analysis and DNA sequencing analysis.

## 2. Results

### 2.1. Identification of HMW-GS by SDS-PAGE and 2-DE

Protein profiles from SDS-PAGE, which separates proteins according to the size, showed that there was no band in the profiles of Elpa and Genoveva with the same electrophoretic mobility as the 6.5 subunit found in Bagou (Figure 1). SDS-PAGE analysis showed that HMW-GS 1B × 6.5 has a molecular weight of approximately 90 kDa and therefore it is more mobile than 1B × 6 (95 kDa), with an apparent molecular weight differences between subunits 1B × 6 and 1B × 7.

These protein profiles were reproducible for the self-identification. Most of them formed a single spot in the 2-DE gel (Figure 2). Analysis showed that isoelectric point of subunit 1B × 6.5 was around 8.7, although a prediction based on primary structure was 8.94. Subsequently, genotype Bagou was compared with genotype Elpa with subunit 1B × 6 (Figure 3). This 2-DE profile showed, that subunit 1B × 6.5 is clearly distinguished from subunit 1B × 6, with a lower pI and molecular weight.

### 2.2. Characterisation of HMW-GS Using MALDI-TOF-MS

In this report, mass spectra of peptides extracted from trypsin-digested protein bands of subunit 6.5 (from Bagou genotype) and subunit 6 (from Elpa genotype) were compared (Figure 4).

Overall, 11 and seven signals were matched to tryptic peptides of subunit 6.5 and 6, respectively (Table 1). The peak assignment was confirmed for the most observed peptides by database search of MS/MS data using the MASCOT tool. The analysis of mass spectra showed five peptides shared between two subunits. Four identified peptides were unique for sequence of subunit 6.5 and only one peptide for sequence of subunit 6.

A high level of identity (99%) was found between the coding sequences of 1B × 6.5 and 1B × 6 HMW-GS genes (GenBank© accession no. LT626205.1 and no. KX454509.1, respectively). The sequence alignment of these subunits is shown in Figure 5. Also high level of identity with other *Glu-1B* alleles were found, e.g., with 6.1 (GenBank© accession no.HQ731653.1, 99%) and 7 (GenBank© accession no. BK006773.1, 93%). Several differences were found between 1B × 6.5 and 1B × 6 sequences. The sequence of 1B × 6.5 subunit has 3 nucleotides less than the sequence of 1B × 6 subunit. There are also 22 single-base substitutions and one three-base deletion.

The HMW-GS 6.5 and 6 sequences are 824 amino acids long and predicted amino acid composition of N- and C-terminal domains of these subunits are identical. The overall identity of the two amino acid sequences is 97.9%. A total of 16 one-amino acid substitutions and one single amino acid deletion were found by a mutual comparison (Figure 6). Accordingly, the predicted molecular weight of 1B × 6.5 subunit (88,322.83 Da) is slightly lower than that for subunit 1B×6 (88,633.09 Da). The novel x-type HMW glutenin subunit 1B × 6.5 was first identified in common wheat https://www.ncbi.nlm.nih.gov/nuccore/LT626205.1 by Gregova et al. (accessed on 7 February 2014).

The HMW-GS 6.5 and 6 sequences are 824 amino acids long and predicted amino acid composition of N- and C-terminal domains of these subunits are identical. The overall similarity of the two amino acid sequences is 97.9%. A total of 16 one-amino acid substitutions and one single amino acid deletion were found by a mutual comparison (Figure 6). Accordingly, the predicted molecular weight of 1B × 6.5 subunit (88,322.83 Da) is slightly lower than that for subunit 1B × 6 (88,633.09 Da).

## 3. Discussion

In last few decades, cereal breeding programs have been focused mainly on the quality and quantity of products, which has caused a reduction of the polymorphism of breeding genotypes. Therefore, landraces and old genotypes with interesting properties are aimed to be involved into process of hybridization to find out new high molecular weight glutenin subunits, which are associated with gluten strength and which could be incorporated into the genomes of current commercial wheat [9,10].

In this work, electrophoretic and MALDI TOF MS analysis of wheat storage proteins indicated novel HMW-GS at *Glu-1B* locus. The HMW-GS are encoded by genes *Glu-1A, Glu-1B* and *Glu-1D*, where numerous alleles were identified [11] and updated. Generally, *Glu-1B* locus is considered as the most polymorphic when comparing to the other loci [12,13]. It is known that the action of HMW-GSs in controlling wheat end-use quality is *Glu-1D* > *Glu-1B* > *Glu-1A*. The polyploidy nature of bread wheat allows for the manipulation of genes on its three sets of homologous chromosomes. Electrophoretic analyses of wheat glutenins is a possible way to detect new HMW-GS alleles [13,14,15]. The protein profiles detected by SDS-PAGE were also analyzed by two-dimensional electrophoresis (2-DE). For this separation IPG strips with 3–10 pH range were used because of wide variation of p*I*s of storage proteins as reported in previous studies [16,17].

With the development of various mass spectrometry techniques, some investigations have focused on measuring the molecular weight of intact HMW-GS [18,19,20,21] despite large molecular weight differences between MS measurements and the gene sequence-based data. Subsequently some authors [16,22,23,24] reported the tryptic peptide mapping of high molecular weight glutenin subunits.

One report [24] concluded that distinguishing between the two sequences comparing the MS sequencing data would hardly be possible because of quantity of identical peptides originated from tryptic cleavages. However, in this case, it is possible to identify three peptide masses referring to the same position of digestion in both subunits, which confirm two single amino acid substitutions between the sequences of 1B×6.5 and 1B×6. The first of these substitutions is located at position 24, where the glutamic acid from subunit 6 is replaced by lysine in subunit 6.5. The second substitution was at the position 819 with differential threonine (subunit 6) and alanine (subunit 6.5). In this report, peptides covering the position 819 were observed in mass spectra (Table 1). The peptides at position 25–33 and 25–36 were identified only for subunit 1B × 6.5 as a result of substituted lysine at position 24 in 1B × 6. In addition, the peptide covering residues 504–518 was identified in 1B × 6. This peptide contains arginine replaced by glycine in 1B × 6.5.

## 4. Materials and Methods

### 4.1. Plant Material

The European wheat genotypes, Bagou (France), Elpa (Germany) and Genoveva (Slovakia), used in this study were obtained from the collection of wheat genetic resources stored in the Gene Bank of the Slovak Republic (National Agriculture and Food Centre, Research Institute of Plant Production, Piešťany, Slovak Republic).

### 4.2. Glutenin Preparation and SDS-PAGE Analysis

The seed storage proteins were extracted from mature kernels or from a part of the kernel without an embryo. The glutenins were extracted, separated by electrophoresis, and visualized according to the International Seed Testing Association standard procedure for SDS-PAGE [25]. SDS-PAGE was performed using 10% acrylamide concentration and Protean II apparatus (Bio-Rad, Hercules, CA, USA) at 30 mA for 6–10 h and a constant temperature 10 °C. Molecular weight standards, Precision Plus Protein^TM^ Standards (Bio-Rad) and the HMW-GSs 1B × 6 (cv. Elpa) and 1B × 7 (cv. Genoveva) were used as the molecular weight markers in the electrophoretic mobility evaluation of the novel 1B × 6.5 et al. subunit expressed by the genotype Bagou.

### 4.3. Protein Extraction and Two-Dimensional Electrophoresis

Storage proteins were extracted also from mature kernels using [26] protocol with some modifications. The samples dissolved in lysis buffer were taken in such concentration which reached 0.1–2.5 mg.mL^−1^ for 2-DE. The Ready Strip™ IPG Strip 17 cm (pH 3–10, Bio-Rad) was placed on it and this assembly was allowed to rehydrate passively overnight. The focusing conditions were: step 1–500 V, step 2–1000 V, step 3–4000 V, step 4–8000 V. The reduced and alkylated strips were washed with 1 × SDS buffer. These strips were then loaded onto 10% SDS-PAGE without any stacking gel. This assembly was sealed using 1% agarose sealing buffer. The gels were run, stained and destained just as for 1-D electrophoresis. The gels were scanned using GS-800™ Calibrated Imaging Densitometer (Bio-Rad).

### 4.4. Protein Identification by Mass Spectrometry (MS) 

Selected bands from 1D SDS-PAGE were excised and processed in accordance with [27] protocol and followed by in-gel digestion with modified trypsin [28]. Tryptic peptides were separated using a simple microgradient device for reversed phase liquid chromatography [29,30]. The microcolumn was first wetted with 5μLof 80% ACN/0.1% TFA (*v*/*v*) and then equilibrated with 0.1% TFA. The peptides were loaded on the system and eluted with gradually increasing ACN content (*v*/*v*) (2 μLof 2% ACN/0.1% TFA, 8% ACN/0.1% TFA, 16% ACN/0.1% TFA, 24% ACN/0.1% TFA, 32% ACN/0.1% TFA and 40% ACN/0.1% TFA). The eluate was directly deposited onto an AnchorChip^TM^ 800–384 target plate in 0.5-μL aliquots and mixed with 0.5-μL of α-cyano-4-hydroxycinnamic acid (Bruker Daltonik, Bremen, Germany). Separated peptides were analyzed with an ultrafleXtreme^TM^MALDI-TOF-TOF-mass spectrometer (Bruker Daltonik) equipped with a LIFT cell and 2 kHz Smartbeam^TM^ II laser (Bruker Daltonik). Mass spectra were obtained in the reflecton positive ion mode with the same instrumental setup, parameters, matrix composition and peptide standards as described by authors [31]. The raw data were processed with DataAnalysis v4.2 SP4 (Bruker Daltonik) and R package MALDIquant 1.19.3 to obtain a list of precursors and corresponding fragmentation data in MGF formatted file. The MGFs were searched against an in-house prepared database containing predicted sequences of glutenins 1B × 6.5 and 1B × 6 supplemented with common contaminants (cRAP protein sequences, The Global Proteome Machine) using Mascot Server 2.5 (Matrix Science, London, UK). Mass tolerances for precursors and fragments ions were set up at ±50 ppm and ±0.5 Da, respectively. Trypsin was set as a protease with 2 missed cleavage allowed; carbamidomethylation of cysteine was set as a fixed modification, methionine oxidation as a variable modification and peptide charge was set at +1. Mass spectra were analyzed in mMass tool [32]. The following peak picking parameters were used to generate mass list: S/N threshold 12, apply baseline, smoothing and deisotoping. Sequence editor implemented in mMass was used to match the spectra with in-silico trypsin digested protein allowing up to 2 missed cleavages and using carbamidomethylation of cysteine as a fixed modification, methionine oxidation as a variable modification and 50 ppm mass tolerance.

### 4.5. DNA Cloning and Sequencing

DNA was extracted from young leaves according to the protocol by previously reported [33]. The quality and concentration was verified electrophoretically and spectrophotometrically. PCRs with primers designed on the base of Glu-1Bx sequence available in public databases were run in Labcycler (Sensoquest, Göttingen, Germany). PCR products were cleaned using High Pure PCR Product Purification Kit (Roche, Mannheim, Germany) and cloned into the pCR^®^4-TOPO plasmid. The resulting ligation products were used to transform *Escherichia coli* TOP10 competent cells according to the manufacturer´s protocol (TOPO^®^ TA Cloning Kit, Invitrogen, Paisley, UK). Purification of plasmids was carried out using High Pure Plasmid Isolation Kit (Roche). Inserts were sequenced using Big Dye 3.1 Sequencing Kit (Applied Biosystems, Foster City, CA, USA) and M13+ and M13-primers. Extension products were separated on an ABI PRISM 3130 sequencer (Life Technologies). Sequences were then treated using T-Coffee [34] software and BLAST analysis available on EMBL web page http://www.ebi.ac.uk/ena/data/sequence/search (accessed on 5 Octorber 2021).

## 5. Conclusions

High molecular weight glutenin subunits of wheat determineits unique dough properties and also baking performance. With the development in proteomics, some authors [7,24] have identified new HMW-GS using electrophoretic and mass spectrometry analysis, but other authors [35,36] also reported novel HMW-GS using mostly DNA analysis. In this study, novel HMW-GS was identified using the most popular electrophoretic methods such as SDS-PAGE and 2D-PAGE. Their combination with MALDI-TOF increased validity of the identification procedure. This report of novel HMW-GS at the *Glu-1B* locus can increase the genetic variability after its transfer into common wheat genetic resources.

## Figures and Tables

**Figure 1 plants-10-02108-f001:**
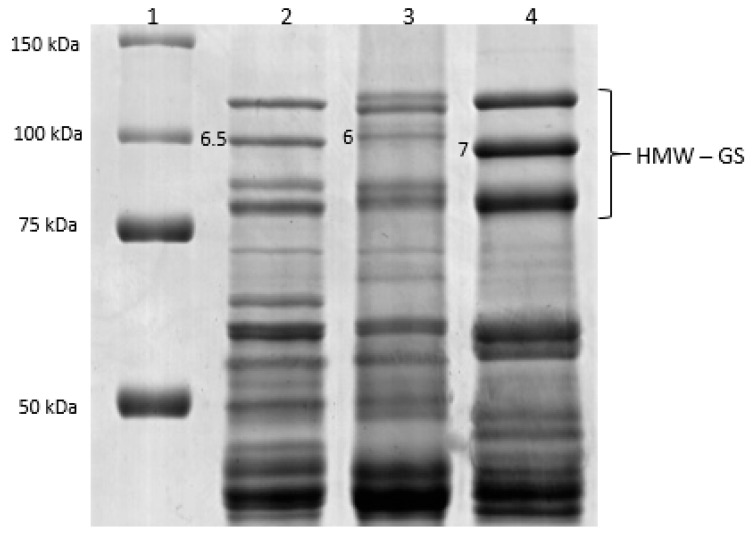
Reduced and alkylated glutenin subunit patterns of selected wheat genotypes analyzed by SDS-PAGE. The HMW-GS were labeled according to the nomenclature of Payne and Lawrence.Lane 1: Precision Plus Protein^TM^Standards (BioRad), lane 2: Bagou genotype, line 3: Elpa genotype, lane 4: Genoveva genotype.

**Figure 2 plants-10-02108-f002:**
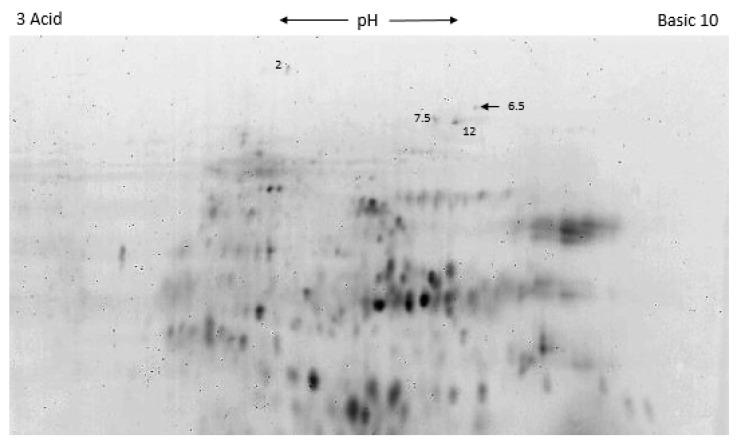
Two dimensional electrophoresis pattern (IEF × SDS-PAGE) of the HMW-GS of wheat genotype Bagou. The arrowhead points to the Glu-1B × 6.5.

**Figure 3 plants-10-02108-f003:**
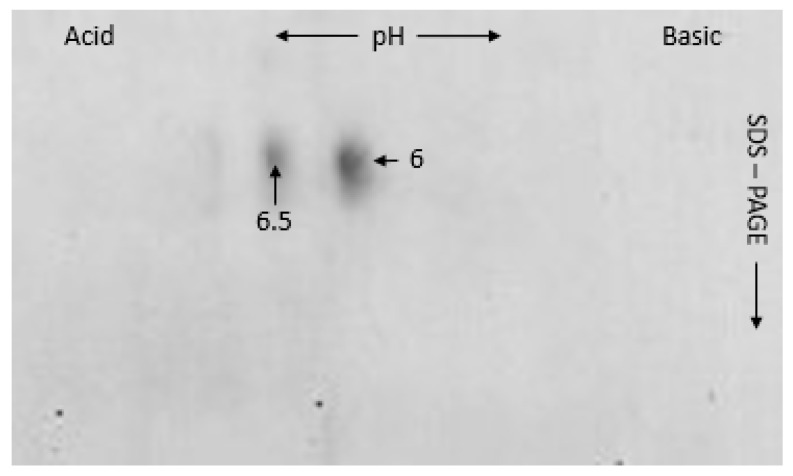
Overlap of two-dimensional pattern of HMW-GS from Bagou and other extract (made with genotype Elpa); an equal mixture. The subunits 6 and 6.5 are indicated.

**Figure 4 plants-10-02108-f004:**
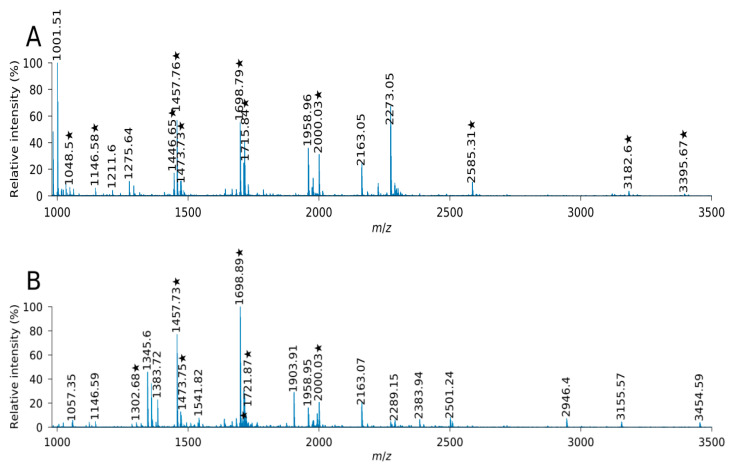
MALDI-TOF mass spectra of the two high molecular weight glutenin subunits 1B × 6.5 (**A**) and 1B × 6 (**B**). Peaks matching peptides of glutenin subunits are marked with asterisk.

**Figure 5 plants-10-02108-f005:**
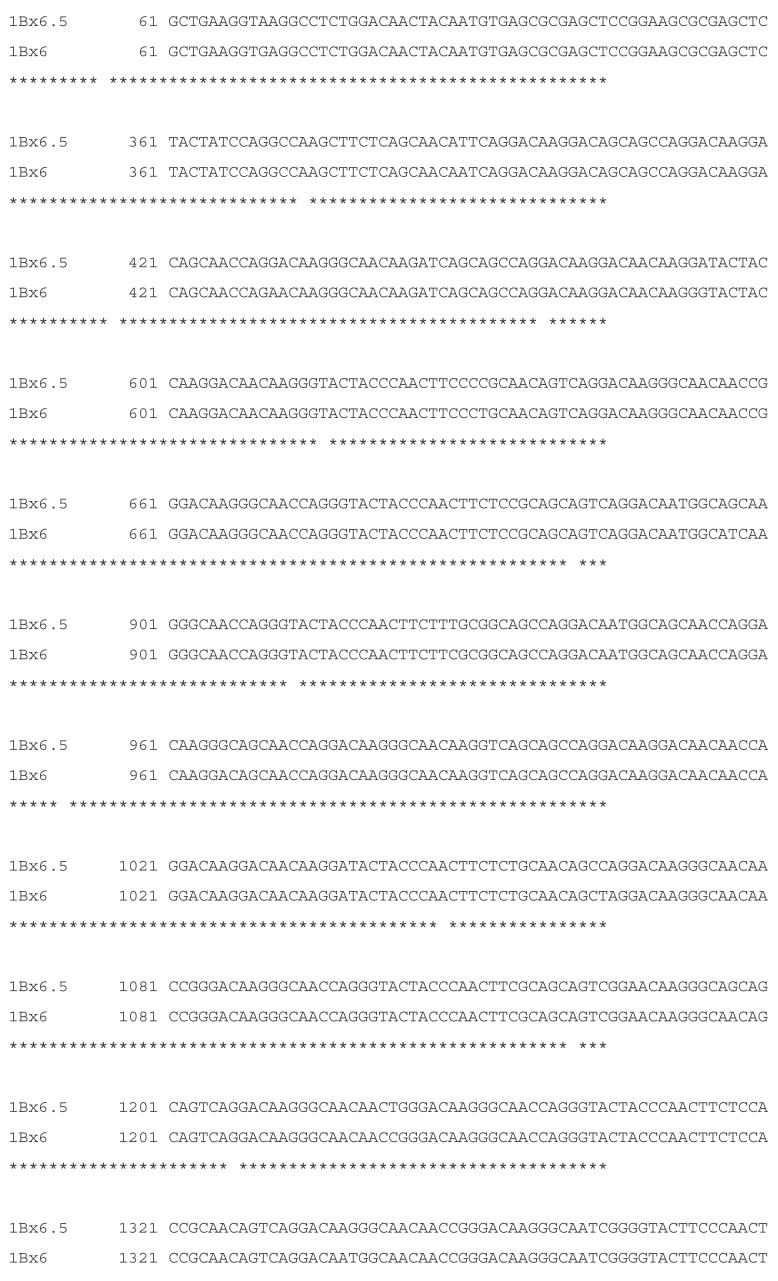
Alignment of 1B × 6.5 (GenBank© accession no. LT626205.1) and 1B × 6 (GenBank© accession no.KX454509.1) HMW-GS nucleotide coding sequences. The similarity between the two sequences is 99% (BLAST).

**Figure 6 plants-10-02108-f006:**
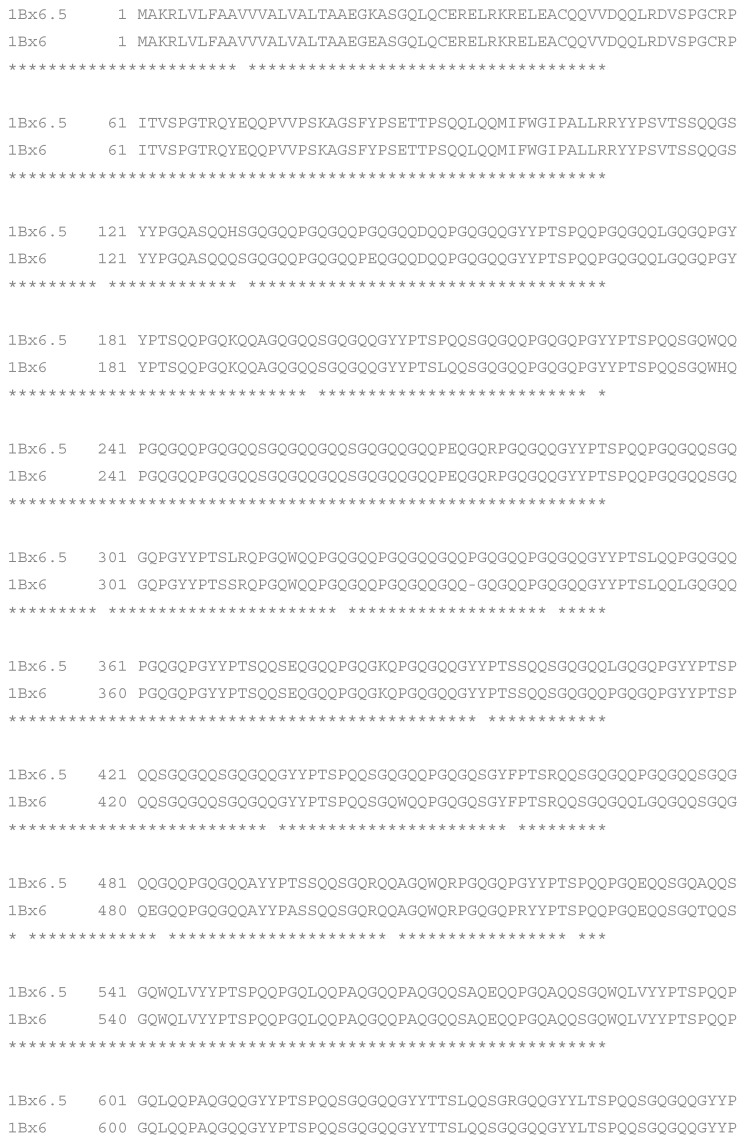
Alignment of the predicted primary structure of 1B × 6.5 and 1B × 6 HMW-GS. The identity between the two sequences is 97.9% (BLAST).

**Table 1 plants-10-02108-t001:** Peptide list of subunits 6.5 and 6 identified by MALDI TOF/TOF MS. Peptides observed in the mass spectra but not confirmed by tandem MS are marked with asterisk. Modifications: carbamidomethylation of cysteine (carb.) and oxidation of methionine (ox.).

Sequence Position in Subunit:		Peptide Mass	Peptide Sequence	Peptide	Peptide Mascot Score:	
1 × B6.5	1B × 6	(Da)		Modification	1 × B6.5	1B × 6
25–33	-	1047.48	ASGQLQCER	carb.	40	-
814–824	-	1145.57	LEGSDALSARQ		38	-
69–79	69–79	1301.66	QYEQQPVVPSK		-	80
25–36	-	1445.7	ASGQLQCERELR	carb.	31	-
801–813	800–812	1456.73	AQQLAAQLPAMCR	carb.	79	86.0
801–813	800–812	1472.72	AQQLAAQLPAMCR	carb., ox.	∗	∗
53–68	53–68	1697.85	DVSPGCRPITVSPGTR	carb.	39	∗
39–52	39–52	1714.83	ELEACQQVVDQQLR	carb.	114	129.0
-	504–518	1720.85	QQAGQWQRPGQGQPR		-	40.0
37–52	37–52	1999.03	KRELEACQQVVDQQLR	carb.	86	82.0
801–824	-	2584.28	AQQLAAQLPAMCRLEGSDALSARQ	carb.	60	-
80–107	80–107	3181.59	AGSFYPSETTPSQQLQQMIFWGIPALLR	ox.	70	-
39–68	39–68	3394.67	ELEACQQVVDQQLRDVSPGCRPITVSPGTR	2 carb.	27	-

## Data Availability

Excluded.

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
