# Peer review of "Molecular Characterization of Novel x-Type HMW Glutenin Subunit 1B × 6.5 in Wheat"

_plants, 2021, doi:10.3390/plants10102108_

Round 1

Reviewer 1 Report

Can you discuss this locus in polyploid perspective. Which of the subgenome is important. Are there any homeologous interactions for this. 

Author Response

Thank you very much for correcting our scientific manuscript. We have accepted all comments and I am sending a corrected scientific manuscript. Please see the attachment.

Reviewer 2 Report

The manuscript described the identification of a novel subunit 1Bx6.5 for Glu-1B, using PAGE electrophoresis (SDS-PAGE and 2D-PAGE) in combination with MALDI-TOF. The responsible nucleotide was cloned and the gene-specific marker was developed. The manuscript is well-written and the results are well-presented. I would recommend publication after minor revision as indicated below:

Abstract: “A high level of identity”, I assume you mean “A high level of similarity”? If yes, please also change the word in other places of the manuscript.

Line 101 Figure 5: Please indicate if this GenBank accession LT626205.1 is a new sequence you submitted? If not, then it means this sequence has been identified before. Has it been published (What is the publication)? If it is not published research, then it should be at least mentioned in this manuscript what was the original research topic (should be available on the gene information page) that resulted in this sequence submission in the first place.

Line 194 Figure 6 “Alignment of the predicted primary structure”: I can only see protein sequence alignment in this Figure, no structure information is given. If it is the results from a structure-based sequence alignment software, please indicate the software name in the figure legend. Again, please change “identity” into “similarity”.

Discussion: Line 267 please change “like at authors” into something like “as reported in previous studies”; Line 274 “Authors” please change into “ One report”.

Materials and Methods: Line 348 please change “by authors” into “previously reported”. Lines 360-361, please indicate if this new sequence has been submitted to the online database or if this sequence was from a previously published or unpublished research (and the research topic of the original sequence submission).

Author Response

Thank you very much for correcting our scientific manuscript. We have accepted all comments and am sending a corrected scientific manuscript. Please see the attachment.
